# Construction of Multiscale Genome-Scale Metabolic Models: Frameworks and Challenges

**DOI:** 10.3390/biom12050721

**Published:** 2022-05-19

**Authors:** Xinyu Bi, Yanfeng Liu, Jianghua Li, Guocheng Du, Xueqin Lv, Long Liu

**Affiliations:** 1Key Laboratory of Carbohydrate Chemistry and Biotechnology, Ministry of Education, Jiangnan University, Wuxi 214122, China; bixinyu@stu.jiangnan.edu.cn (X.B.); yanfengliu@jiangnan.edu.cn (Y.L.); lijianghua@jiangnan.edu.cn (J.L.); gcdu@jiangnan.edu.cn (G.D.); lvxueqin@jiangnan.edu.cn (X.L.); 2Science Center for Future Foods, Ministry of Education, Jiangnan University, Wuxi 214122, China

**Keywords:** multiscale genome-scale metabolic models, multiconstraint models, multiomics models, machine learning, whole-cell models

## Abstract

Genome-scale metabolic models (GEMs) are effective tools for metabolic engineering and have been widely used to guide cell metabolic regulation. However, the single gene–protein-reaction data type in GEMs limits the understanding of biological complexity. As a result, multiscale models that add constraints or integrate omics data based on GEMs have been developed to more accurately predict phenotype from genotype. This review summarized the recent advances in the development of multiscale GEMs, including multiconstraint, multiomic, and whole-cell models, and outlined machine learning applications in GEM construction. This review focused on the frameworks, toolkits, and algorithms for constructing multiscale GEMs. The challenges and perspectives of multiscale GEM development are also discussed.

## 1. Introduction

Genome-scale metabolic models (GEMs) transform cell growth and metabolism processes into a mathematical model based on a stoichiometric matrix and solve the optimal solution of the target equation at a steady-state [1]. GEMs have become an important tool for systematically revealing cell growth and metabolic regulation [2]. To satisfy the needs of different growth and metabolism processes in actual cells, researchers have developed a framework for different constraint models and various model analysis algorithms. Thus, GEMs are widely used in guiding strain design, predicting cell phenotype, analyzing metabolic mechanisms, mining unknown metabolic pathways, and studying the evolution of strains [3].

Since the first GEM of *Haemophilus influenzae* RD was reported in 1999 [4], various GEMs have been constructed for 5897 bacteria with the development of genome sequencing and omics analysis techniques [5]. In particular, many GEMs have been constructed for classical industrial microorganisms, such as *Escherichia coli* [6], *Saccharomyces cerevisiae* [7], and *Bacillus subtilis* [8]. The first GEM of *E. coli*, one of the most important model organisms, was reported in 2000 [9]. Thirteen GEMs have been reported, with four updates in gene–protein response correlation coverage and prediction accuracy [10]. In the latest GEM of *E. coli*, the metabolism-expression (ME) model was reported, which reconstructs the complete pathway of transcription and translation during cellular metabolism [11]. FoldME [12], OxidizeME [13], and AcidifyME [14] were developed based on an ME model to simulate different environmental pressures on temperature, oxidation and low pH, respectively. *S. cerevisiae* was the first eukaryotic microorganism whose genome was sequenced [15]. Thus far, 13 GEMs for *S. cerevisiae* have been reported, and the latest Yeast8 can dissect the metabolic mechanism of cells at multiscale levels [7]. In *B. subtilis*, seven GEMs have been reported. The latest ec*i*YO844 integrates the enzymatic data of central carbon metabolism to guide the design of high-producing poly-γ-glutamic acid strains [16].

To explore the relationship between the genotype and phenotype in cells, flux balance analysis (FBA) is widely used to characterize cellular metabolism [17]. Then, dynamic flux balance analysis algorithms, dynamic FBA, were developed to meet the design of model chassis cells [18]. However, FBA suffers from the limitations of assuming a steady-state and substrate uptake rate as constraints. Therefore, model analysis algorithms based on multi-omics data were developed to improve the application scope of GEMs, such as iMAT [19], MADE [20], ΔFBA [21], GIM3E [22], multiTFA [23], and INTEGRATE [24]. In addition, the development of various omics databases and model-building tools facilitates the construction of multi-scale GEMs, such as KBASE [25], ModelSEED [26], CarveMe [27], and MEMOTE [28]. With the development of high-throughput technologies, massive omics data drive the interpretation of biological mechanisms [29,30]. In particular, machine learning has become an indispensable tool for the training and analysis of large datasets [31]. Therefore, many machine learning-trained GEMs that integrate multilevel omics data to deepen insights into genotype–phenotype relationships have been reported [32].

However, the single gene–protein-reaction relationship in GEM often leads to mispredictions due to the multifactorial regulation of microbial metabolism. Therefore, multiscale GEMs that add constraints, such as thermodynamic, enzymatic, or kinetic constraints, or integrate omics data, such as proteomic, transcriptomic, or other omics data, have been developed based on traditional GEMs, and have been widely used in silico biodesign. This review summarized the construction workflow and toolkits of multiscale models. It also discussed how to use artificial intelligence, such as machine learning, to improve the qualities of multiscale GEMs. Finally, this review analyzed the challenges and perspectives of multiscale GEM development in the future. This review may aid biological engineers in the in silico design of versatile cell factories for sustainable bioproduction.

## 2. Constraint-Based GEMs

GEMs have been widely used to simulate metabolic phenotypes at the systems level, often relying only on constraints on metabolite uptake rates. However, cellular metabolism, a fundamental biological process used by all organisms to generate and consume energy to promote growth, not only depends on the regulation of interconnected mechanisms within cells, but is also affected by the external environment. Multiple factors regulate cellular metabolism for an organism to respond to various conditions. Therefore, the basic form of GEMs cannot explain the complex regulatory mechanisms within the cell. This limitation of GEMs has prompted the development of multiple constraints to integrate regulatory mechanisms, thereby improving the predictive power and broadening the scope of GEMs. Several constraint-based models have been developed, including thermodynamic, enzymatic, and kinetic constraint models (Figure 1).

### 2.1. Thermodynamic Constraint GEMs

Although classic GEMs can achieve flux analysis of metabolic networks, single stoichiometric and metabolite concentration limitations hinder their scope of application [17]. Therefore, introducing thermodynamic constraints can narrow the range of feasible solutions by considering the directionality and Gibbs free energy of metabolic reactions [33].

The development of thermodynamic constraints relies on three main algorithms: energy balance analysis (EBA), network embedded thermodynamic analysis (NET analysis), and thermodynamically based metabolic flux analysis (TMFA; Table 1). EBA provides additional constraints for the metabolic network based on voltage loop laws and effectively reduces the feasible flux space compared to FBA [34]. NET analysis was proposed as a computational thermodynamics-based framework that couples quantitative metabolomic data into metabolic networks via thermodynamic laws and Gibbs free energies of metabolites. NET analysis enables the identification of putative genetically or allosterically regulated active sites and can be used to explore new interrelationships in metabolic regulation [35]. Henry et al. proposed that TMFA, which uses mixed-integer linear constraints to generate flux analysis, and the flux distribution produced by TMFA, eliminates any thermodynamically infeasible reactions and pathways [36]. TMFA first introduced linear thermodynamic constraints into GEMs, pioneering the construction and analysis of models based on thermodynamic constraints.

However, the standard Gibbs free energy of metabolites is mostly unknown, and the temperature, pH, and ionic strength of different cells can have huge effects on the detection of Gibbs free energy. To overcome this challenge, Mavrovouniotis proposed a method for estimating Gibbs free energy and the equilibrium constants of biochemical reactions by multiple linear regression from group contributions [37]. In the group contribution method, the molecular structure of a single metabolite is decomposed into a set of linear molecular substructures based on structural assumptions, and its linear model can more conveniently estimate the Gibbs free energy of metabolite formation and metabolic reactions [37]. The eQuilibrator website was developed to obtain online biochemical equilibrium constants and Gibbs free energies of metabolites and metabolic pathways [38]. Various algorithms and toolkits were developed for model construction and analysis, such as OptMDFpathway (an algorithm for directly calculating thermodynamic driving forces in metabolic pathways) [39], Find_tfSBP (an algorithm for identifying thermodynamically feasible minimal equilibrium pathways for high-yielding target products in metabolic networks) [40], matTFA, and pyTFA (a toolkit for integrating thermodynamic data with constraint-based GEM) [41].

Based on the above algorithms and frameworks, many efforts have been made to explore the construction and analysis of thermodynamically constrained models. The first thermodynamically constrained model in *E. coli*, iHJ873, evaluated the thermodynamic feasibility of the reactions in the model through Gibbs free energy, focused on the thermodynamic study of a single reaction, and explored the flux direction of the reaction [42]. Gibbs free energy change values for 1403 (97%) reactions estimated by the group contribution method were included in iBsu1103 of *B. subtilis*, identifying 653 (45%) irreversible reactions in the model, bringing its prediction accuracy from 89.7% to 93.1% [43]. In addition to determining the direction of reaction fluxes and assessing the thermodynamic feasibility of metabolic reactions in the model, thermodynamics is applied to metabolite sensitivity analysis, which combines constrained modeling, design of experiments, and global sensitivity analysis to evaluate metabolites in the model [44]. The quantitative relationship between the regulation of metabolic flux by enzymes and thermodynamics in the metabolic network was explored, and the thermodynamic driving force of the network constrains almost all flux control coefficients in the pathway [43]. The effects of thermodynamic constraints on the prediction of metabolic networks were evaluated, and the networks with thermodynamic constraints effectively improved the prediction accuracy of essential genes [45]. These studies comprehensively highlight the importance of global thermodynamic signatures in limiting metabolic regulation patterns.

### 2.2. Enzymatic Constraint GEMs

Models based on stoichiometric relationships and thermodynamic constraints have been widely used to predict cell growth rates, explore the interactions of metabolic pathways, and identify potential targets for metabolic engineering. However, the limitations of substrate uptake rates and the thermodynamic feasibility of metabolic reactions are insufficient to describe complex metabolic networks in which enzyme kinetics are a nonnegligible factor in regulating cellular metabolism.

Four frameworks or toolkits can be used for constructing enzymatic constraint GEMs. (1) FBA with molecular crowding (FBAwMC) limits the concentration of enzymes that catalyze various metabolic reactions in the crowded cytoplasm, and each enzyme can solve the crowding factor based on six parameters (molecular weight, mass volume, Km, *kcat*, substrate concentration, and cytoplasmic density) [46]. (2) Metabolic modeling with enzyme kinetics (MOMENT) predicts metabolic flux and growth rates using enzyme turnover rates and molecular weight. Importantly, it considers specific enzyme concentration requirements for catalyzing predicted metabolic flux rates, including isozymes, protein complexes, and multifunctional enzymes [47]. (3) A comprehensive modeling framework, GEMs with enzymatic constraints using kinetic and omics data (GECKO), limits metabolic flux in GEMs based on enzyme kinetics and protein abundance [43]. In GECKO, each metabolic reaction was split into putative reactions catalyzed by an enzyme, and each putative reaction is limited by the abundance of that enzyme [48]. This allows the direct integration of quantitative proteomic data, significantly reducing model flux variability in metabolic reactions. (4) The AutoPACMEN toolbox automates the creation of enzymatic constraint models, especially the automatic reading and processing of enzymatic data from different databases [81]. It simplifies the construction and analysis of enzyme constraint models and paves the way for the routine construction of enzyme constraint models for different strains.

Aside from the four toolkits, several algorithms have been developed to introduce enzymatic constraints in GEMs. Integrative omics–metabolic analysis quantitatively integrates proteomic and metabolomic data with GEMs, taking into account the concentration levels of enzyme substrates and products to predict metabolic flux distributions more accurately [82]. Enzyme cost minimization calculates the number of enzymes for metabolic flux at the lowest protein cost by prior distributions, thermodynamic laws, and Bayesian statistics [83]. The proteome allocation theory divides the entire proteome into three modules (fermentation, respiration, and cellular activity) and explores the effects of cellular energy demand on overflow metabolism [84].

In *E. coli*, a model of metabolic flux balance was constructed based on FBAwMC, which can activate cellular metabolism by systematically recognizing environmental changes [46]. Vazquez et al. demonstrated the effects of limited solvent capacity on the growth rate of cells and explored a regulatory mechanism that identifies metabolic control switches in the central carbon cycle by FBAwMC [85]. Furthermore, Adadi et al. demonstrated that, compared to FBAwMC, the model constructed by MOMENT could significantly improve the prediction accuracy of various metabolic phenotypes by conducting growth experiments in a minimal medium with 24 single carbon sources [47]. However, the assumption in the MOMENT algorithm that enzymes are in a substrate-saturated state does not conform to the actual cell growth state. Hence, the upper limit of each enzyme usage in ecYeast7 of *S. cerevisiae* was defined at the protein level, and the expected constraints of each flux were specifically considered by GECKO [48]. Moreover, ecYeast7 cannot only accurately simulate the maximum specific growth rate of cells under different carbon sources and reduce the flux variability of the model, but also explain the physiological reactions of cells, such as overflow metabolism and cell adaptation under temperature stress through the enzyme restriction theory.

Based on GECKO, the protein requirements for lysine synthesis were predicted by ec_iML1515 (enzyme-constrained model for *E. coli*), and the expression of 20 proteins was optimized by modular engineering, resulting in a lysine titer of 193.6 ± 1.8 g/L, which increased by 55.8% [86]. Model ec-iBag597 (enzyme-constrained model for *Bacillus coagulans*) estimated the protein efficiency of major ATP-producing pathways in cells, paving the way for a comprehensive understanding of *B. coagulans* metabolism [87]. Model ec*i*JB1325 (enzyme-constrained model for *Aspergillus niger*) predicts the differential expression of enzymes under different growth conditions and significantly reduces the solution space of the model by 40.10%, explaining the changes in metabolic phenotypes at the proteomic level [88].

### 2.3. Kinetic Constraint GEMs

Although enzyme constraint GEMs have been widely used in metabolic engineering, the enzyme parameters set in the hypothetical homeostasis are not suitable for the dynamic growth of cells in complex environments. In contrast, kinetic constraint GEMs enable dynamic analysis of biological systems and can overcome the shortcomings of traditional models. Moreover, kinetic constraint GEMs estimate reaction rate rules from metabolic phenotypes and can capture the effects of fluctuations in enzyme activity on metabolic flux.

After the central carbon metabolism kinetic model of *E. coli* was constructed in 2002 [89], researchers started to explore the modeling framework of the kinetic model, and five toolkits have been developed as follows: (1) Structural kinetic modeling (SKM) was developed based on the Jacobian matrix (the matrix captures the dynamic response of the metabolic system), where the matrix can construct a dynamic linear approximation of the metabolic system in the absence of dynamic data. It enables SKM to perform dynamic analysis of metabolic systems with minimal data, providing a versatile framework for exploring possible system dynamics [49]. (2) The mass action stoichiometric simulation (MASS) framework defines the Jacobian matrix of a biochemical reaction network as a product of an S matrix and a G matrix, where the S matrix is the stoichiometric matrix, and the G matrix is composed of fluxomics and metabolomics data, and also performs the kinetic characterization and thermodynamic evaluation of each reaction. The MASS framework enables the assessment of kinetic (k PERC) and dynamic (Jacobian) properties of large metabolic systems to formulate time-scale hierarchies in biological networks, which are the most scalable dynamics, by combining network topology and multiomics data learning model frameworks [50]. (3) Optimization and risk analysis of complex living entities (ORACLE) uses Monte Carlo sampling to calculate the elasticity distribution of enzymes in uncertain states based on the MCA framework and fully considers the enzyme state space to determine the effects of enzyme-regulatory interactions on metabolic networks [51]. ORACLE obtains a population of control coefficients consisting of Jacobian and elastic parameters to accurately characterize the dynamic state of a metabolic system by integrating network structure with fluxomics data supported by directionality based on thermodynamic and metabolomic data. Notably, ORACLE captures the global properties of metabolic networks, identifies control features in any given network, and determines the probability distribution of control coefficients for different network configurations (represented by ensemble entities) [52]. (4) Ensemble modeling (EM) develops an ensemble of steady-state kinetic models based on an iterative process of determining kinetic parameters based on reaction reversibility and enzyme distribution. EM predicts different phenotypes with dynamic responses by constructing a set of initial models with different kinetic behaviors and trains the models on the acquired phenotypic data to determine the smallest kinetic model. Notably, for unknown enzyme kinetics, EM resolves the enzymatic reaction by mass action kinetics to capture the saturation behavior and substrate-level regulation of the reaction [53]. (5) Approximate Bayesian computation-general reaction assembly and sampling platform (ABC-GRASP) parameterizes the data sampled in GRASP and uses ABC to calculate the data, providing a framework for dissecting the mechanism of enzyme-catalyzed reactions through kinetic information under uncertainty [90]. However, in all kinetic frameworks, ABC-GRASP requires more experimental data to reveal the effects of thermodynamic affinity, substrate saturation level, and effector concentration on flux control and response coefficients of various enzymatic reactions [91].

Based on the five toolkits, various algorithms have been developed and applied to the construction and analysis of kinetic models. EM for robustness analysis (EMRA) was developed based on numerical continuation and EM to investigate the robustness of unnatural metabolic pathways. The bifurcation robustness of the two synthetically central metabolic pathways (nonoxidative glycolysis and the reverse glyoxylate cycle) that achieve carbon conservation was analyzed by EMRA, weighing robustness and performance in the regulation of metabolic flux [92]. An in silico approach to the characterization and reduction of uncertainty in the kinetic models of genome-scale metabolic networks (iSCHRUNK) was developed based on the ORACLE framework and machine learning to determine and quantify the kinetic parameters of enzymes to obtain more accurate kinetic parameter ranges, thereby reducing the uncertainty of the model [93]. DMPy is proposed as a computational framework to automatically search kinetic rates to generate metabolite fluxes, which can analyze the effects of parameter uncertainty on model kinetics and can be used to test how model simplification changes metabolic system properties [94]. MASS python (MASSpy) was developed as a toolkit for reconstructing, simulating, and visualizing dynamic metabolic models. MASSpy solves data-driven problems in dynamic modeling programs with a combination of constraint-based and kinetic modeling that makes it possible to exploit mass action kinetics and detailed chemical mechanisms to build dynamic models of complex biological processes [95].

Based on the above frameworks and toolkits, a kinetic model for *E. coli*, k-ecoli457, was constructed by combining a genetic algorithm (GA) and EM, and the model was parameterized by minimizing the difference between the model predictions and the steady-state flux distributions of the 25 mutant strains. The prediction results showed that the average relative error of k-ecoli457 for the prediction of 129 product yields in 320 designed strains was within 20% of the measured value, showing the accuracy of k-ecoli457 in predicting the phenotype of *E. coli* under different growth conditions of genetic perturbation [96]. In *B. subtilis*, a kinetic model was developed to describe growth and sporulation as the process of differentiation from vegetative cells to spores. The growth kinetics of spores was described by two specific parameters: time and probability of spore formation. In addition, the biological significance of sporulation parameters was assessed experimentally, qualitatively, and quantitatively at the physiological level of the sporulation process in *B. subtilis* [97]. For *Clostridium thermocellum*, the core kinetic energy model of *C. thermocellum* was constructed based on EM, named k-ctherm118, and the model was parameterized by the fermentation data of 19 metabolites of lactic acid, malic acid, and the hydrogen production pathway so that k-ctherm118 could capture the upregulation of amino acid production and predict the direction and extent of changes in cytosolic concentration under ethanol stress [98].

### 2.4. Multiconstraint GEMs

Although multiple kinetic modeling frameworks have been developed and kinetic models of multiple strains have been constructed to reveal the regulatory mechanisms of metabolic networks, datasets for model parameterization and computational power hinder the development of kinetic constraint models. Therefore, comprehensive GEMs integrating more constraints were developed.

Yang et al. proposed a Pyomo-based model framework integrating enzymatic and thermodynamic constraints and constructed a multiconstrained model for *E. coli* [54]. Moreover, the optimal pathways for 22 metabolite products were calculated, and among the predicted L-arginine synthesis pathways, thermodynamically unfavorable and high enzymatic cost pathways were excluded from achieving an accurate prediction of metabolites [54].

In addition, the most classic example of multiconstraint GEMs is the ME model. The ME model was reported for *E. coli*, which extended the transcription and translation processes in cell growth metabolism based on the traditional GEM (M-model) [11]. Unlike the M-model, the ME model is combined with the M-model and E-matrix through metabolite and coupling constraints. The E-matrix contains 11,991 components and 13,694 biochemical reactions, depicting gene expression and all components and modification processes of protein synthesis in *E. coli* [99]. In addition, the E-matrix contains all gene products necessary to produce the active ingredient and incorporates known reaction stoichiometry, including protein-substrate complex intermediates, metal ions, and cofactors. It also considers the necessary modifications to stabilize RNA and proteins, as well as rRNA and tRNA processing reactions, providing an accurate representation of operons in biology [99]. Thus, compared to constraint-based models, ME models reconstruct the complete pathway of transcription, translation, and metabolism, enabling the simulation of protein composition and the calculation of the cellular cost of enzyme synthesis [100]. Importantly, the ME model accurately decouples the three stages of substrate uptake, growth rate, and growth yield during cell growth and metabolism, allowing for trade-offs between the rates and yields of important products [101].

With the development of the software COBRAme [102], the construction of ME models was quickly extended to other microorganisms. For *Thermotoga maritima*, the ME model was constructed to accurately simulate changes in cell composition and gene expression, in which experimental values of the transcriptome and proteome containing 651 genes were positively correlated with the simulated values, and the Pearson correlation coefficients were 0.54 and 0.57, respectively [103]. For *Clostridium ljungdahlii*, the first ME model of Gram-positive bacteria was reported, covering the synthetic pathways of biomass composition, revealing the influence of protein partition and medium composition on the metabolic pathways and energy conservation of the strain and significantly broadening the model prediction range [104]. In addition, the ME model of *E. coli* has undergone several updates, such as iOL1650-ME (revealing the importance of proteomic constraints for cell growth and secretion of by-products) [11], iJL1678-ME (revealing predictions of perturbations, such as membrane crowding and enzyme efficiency impact) [100], and iJL1678b-ME (reducing free variables and solution time to improve model prediction accuracy) [102]. To address different stress responses in the metabolic environment of cell growth, the ME model integrates with known response mechanisms, extending FoldME (predicting temperature-dependent growth rate and protein abundance changes) [12], OxidizeME (predicting changes in cellular phenotypes under oxidative stress) [13], and AcidifyME (achieving a systemic understanding of acid stress response) [14].

Based on the ME model, Salvy et al. developed a framework for expression and thermodynamics flux models (ETFL), which formulated a mixed-integer linear program (MILP) to integrate metabolites, proteins, and mRNA, enabling the simultaneous consideration of expression, thermodynamics, and growth-related variables [55]. This framework provides finer control and more accurate prediction of gene editing, with fewer false-negatives for ETFL predicting gene necessity in *E. coli* than iJO1366 [55]. Furthermore, yETFL was developed in *S. cerevisiae*, which extends the eukaryotic system (additional ribosomes and RNA polymerase in the eukaryotic mitochondrial expression system) based on ETFL and constrains proteins assigned to metabolism and cellular expression. Therefore, yETFL can capture the Crabtree effect only by integrating experimental data [105,106].

## 3. Multiomics-Integrated GEMs

Although the multiconstraint approach in GEMs allows researchers to explore cellular metabolic networks, there are still certain difficulties in analyzing complex regulatory mechanisms in cells [102]. Therefore, GEMs integrating the transcriptional regulatory network (TRN) and protein structure (PRO) were constructed to comprehensively analyze the regulatory mechanism of the metabolic network in cells and the feedback regulation of metabolic flux at the genome scale to understand the growth and metabolic process of cells in detail (Figure 2).

### 3.1. TRN-Integrated GEMs

Transcriptional regulation is one of the important mechanisms by which microorganisms transform their metabolic flux in response to changing environments. TRNs have been widely reported in bacteria after the standard procedure for reconstituting TRNs was proposed [107]. TRNs usually appear as a network of mutual regulation between genes, and global transcription factors control the expression of most genes.

Two tool platforms for the integration of TRN into GEMs based on logical Boolean rules were developed: the toolbox for integrating genome-scale metabolism (TIGER) [56] and FlexFlux [57]. TIGER converts generalized Boolean and multilevel rules into MILPs and couples these rules into GEMs to address the multiple iterations required to reach a steady-state for multilayered transcriptional regulation compared to traditional single iterations [56]. Unlike TIGER, FlexFlux has a user-friendly graphical interface, and it applies the regulatory steady-state analysis algorithm to constrain each component in the network to a single steady-state [57]. Importantly, FlexFlux allows the transformation of discrete qualitative states of regulatory networks into user-defined continuous intervals and the different approaches to a detailed analysis of regulatory mechanisms in metabolic network models [57].

Furthermore, the probabilistic regulation of metabolism (PROM) [58], gene expression and metabolism integrated for network inference [108], and transcriptional regulation FBA [59] realized the coupling of GEMs and transcriptional regulation models based on Boolean rules and explored the effects of transcription factors on the cell phenotype in different environments. Based on PROM, the integrated deduced and metabolism (IDREAM) method [109] and the optimization of regulatory and metabolic networks approach (OptRAM) [60] were developed to evaluate the regulatory role of transcription factors in metabolic networks. The strategy of optimal gene combination optimization can be inferred to improve the yield of the target product.

For *E. coli*, TRN-integrated GEMs were constructed from quantitative cell growth data [58]. Six strains with key transcriptional regulator knockout in the oxygen consumption reactions were constructed according to the model prediction. The model accurately predicted the growth rate of 14 knockout phenotypes, with a correlation coefficient of 0.95 [58]. For *Mycobacterium tuberculosis*, an expanded knowledge base of metabolic networks and regulatory mechanisms was constructed with 104 TF regulatory networks based on ChIP-seq interactions linked to 810 GEMs [110]. The knowledge base identified synergistic TF–drug interactions in >50% of the cases, suggesting that this model may provide corresponding information for antituberculosis drug target identification [110]. For *S. cerevisiae*, a TRN-integrated GEM was constructed involving 25,000 regulatory interactions and controlling 1597 metabolic reactions [108]. The model accurately predicted the phenotype of TF knockout under different conditions and revealed potential condition-specific regulatory mechanisms [108]. Furthermore, Shen et al. used OptRAM to design efficient synthetic strains of succinic acid, 2,3-butanediol, and ethanol in yeast and confirmed the role of key predicted genes [60]. The productivity of 2,3-butanediol increased by 61 times compared to the experimental value under the optimization strategy simulation, and the productivity of ethanol increased by 1.8 times under the same conditions [60].

### 3.2. PRO-Integrated GEMs

GEM construction relies on the mining of multiomics and the analysis of cellular metabolic processes, in which protein–protein interactions control a wide range of cellular processes, such as signal transduction [110,111] and molecular transport [112]. Therefore, introducing proteomic data into GEMs can provide insights into metabolic network mechanisms in cells [113].

Brunk et al. proposed the GEM with a protein structure (GEM-PRO) framework, which directly maps genes to transcripts, PROs, biochemical responses, network states, and, ultimately, phenotypes [61]. The massive open-source protein database provides >110,000 entries of biological macromolecular structure information [114]. These have facilitated the development of protein ensemble models. Chang et al. integrated GEMs with data such as amino acid sequence, PRO, functional annotation, and protein-substrate binding sites to analyze protein stability in the cellular environment [115]. PRO-integrated GEMs predicted the growth-limiting factor of heat resistance and revealed the metabolic mechanism of heat resistance for *E. coli* [115]. GEM-PROs of *E. coli* and *T. maritima* were reported, revealing growth limitation by protein instability through features such as temperature conditions, protein folding, and substrate specificity [61]. The establishment of this model demonstrates the utility of the intersection of systems biology and structural biology [61].

Recently, an integrated GEM based on protein synthesis and degradation was reported in yeast, which systematically alters the growth rate and determines its protein expression level [116]. Importantly, this model identifies protein compartment-specific constraints to reveal growth rate-optimized protein expression profiles, providing a framework for understanding metabolic mechanisms in eukaryotic cells [116]. However, except for *E. coli* and yeast, PRO-integrated GEMs have not been widely used, and the acquisition of accurate PRO data may be the main limiting factor for its development.

### 3.3. Comprehensive Metabolic Models

Cellular metabolism is regulated at multiple levels, so a single integrated model cannot accurately predict cellular phenotypes under various environmental conditions. Therefore, the development of comprehensive models facilitates the exploration of cellular metabolism at multiscale levels.

In *E. coli*, a comprehensive modeling framework (EcoMAC), which unifies various biological processes and multilayer interactions, was developed to combine gene expression data from genetic and environmental perturbations, transcriptional regulation, signal transduction and metabolic pathways, and growth measurements [117]. In this framework, expression balance analysis was used to integrate genetic, competence, phenomenological, and environmental constraints to predict gene expression, and a new approach to transcription-based metabolic flux enrichment was developed to expand flux boundaries and simultaneously calculate metabolic interaction with transcription [117]. Notably, EcoMAC improved the performance of the region classifier to 22%, identifying stress responses, locomotion and taxis, and cell motility, which were the most abundant biological processes from 500 computationally inferred interactions [117]. A knowledge base calculating the traits of *E. coli*, iML1515, was reported, which contained not only transcriptome, proteome, and metabolome data, but also unique metabolite response information and complete PRO data [118]. The knowledge base simulated 23,617 phenotypic data with 93.4% accuracy in gene knockouts of 16 different carbon sources and identified important metabolic differences in clinical isolates [118]. These all reflect its potential for identifying drug targets and then using them in therapeutic and clinical applications.

For *S. cerevisiae*, a genome-wide tool for multiscale modeling data extraction and representation (GEMMER) was developed. This tool aids the visualization of the physical, regulatory, and genetic interactions between proteins and genes and integrates existing database information to support multiscale modeling efforts [62]. Lu et al. introduced a model ecosystem based on the Yeast8 model platform, which includes ecYeast8 (enzyme constraint model), panYeast8 (protein 3D structure database), and coreYeast8 (core metabolic network model of 1011 different mutant strains of *S. cerevisiae*) [119]. This model ecosystem comprehensively explores the effects of single nucleotide variation on phenotypic characteristics, promotes the exploration of yeast metabolism at the multiscale level, and provides guidance for the wide application of yeast systems and synthetic biology [119].

## 4. Whole-Cell Model

Although various multiscale integrated models have been established to simulate cell growth and metabolism, many subcellular processes have not yet been incorporated, such as chromosome initiation and replication, protein activation and folding, and RNA decay and modification [120]. Therefore, the development of whole-cell models becomes the “ultimate goal” of systems biology.

### 4.1. Construction of Whole-Cell Models

Whole-cell models are computational models that explain the integrated function of every gene and molecule in a cell, aiming to predict the cellular phenotype from the genotype by representing the entire genome, the structure and concentration of each molecular species, each molecular interaction, and the extracellular environment [121].

The construction of whole-cell models can be divided into five stages: (1) Training data. The biological system of cells is divided into functional modules, and the data for the cellular process of each module are collected [121]. These data can be obtained from large databases and a massive amount of literature. Machine learning can automatically rebuild knowledge bases for data sorting and cleaning [63,122,123]. Open-source tools can be used for data training [64,65,124,125]. (2) Submodel integration. Each pathway model is built according to experimental data that can be integrated according to the model database [66,126,127], and undefined pathways or data can be built by relying on rule-based tools, such as E-Cell [67], CellDesigner [68], and COPASI [69]. Next, the hybrid simulator integrates heterogeneous submodels based on simultaneous time steps [70,128,129]. (3) Parameter estimation. After building the structure of the model, the parameters need to be identified to match the model predictions with the experimental data. Due to the high dimensionality and supercomputing requirements of whole-cell models, it is necessary to simplify the model to optimize the parameters [130] and identify the parameters using automatic differentiation, parallelized simulation engines, and distributed optimization procedures [71]. (4) Model refinement. After building the model, massive data are needed to iteratively evaluate the model, in which the prediction of the model phenotype is the focus of validation, requiring multilevel validation of the model’s accuracy. It is a huge challenge to obtain massive experimental data, which can be obtained from microfluidics [131] and high-throughput experiments [132]. (5) Visual analysis. Visualization tools are the best means to analyze complex and multilevel whole-cell models. Many simulation tools have been developed to explore cellular energy metabolism and analyze cell-to-cell interactions, such as WholeCellSimDB [133], WholeCellViz [72], and E-Cell [67].

### 4.2. Application of Whole-Cell Models

Currently, whole-cell models have been constructed for *Mycoplasma genitalium* [134] and explored in *E. coli* [135] and *S. cerevisiae* [136], providing new insights into many previously unobserved cellular behaviors. The first whole-cell model was reported for *M. genitalium*, which describes the life cycle of a single cell at the level of individual molecules and their interactions [134]. A total of 128 wild-type cells were simulated using this model, and predictive simulations included cellular and molecular properties, such as cell mass and growth rate, as well as the count, localization, and activity of each molecule [134]. Results showed that the model calculation was completely consistent with the experimental data regarding doubling time, cytochemical composition, and gene expression [134]. In addition, the model successfully predicted central carbon cycle flux, protein synthesis, and mRNA- and protein-level distribution with high accuracy [134]. Therefore, the model accurately predicts a wide range of observable cellular behaviors. Notably, the establishment of this whole-cell model provides a framework for comprehensive modeling of systems biology in other strains.

For *E. coli*, a large-scale mechanistic model that evaluates large heterogeneous datasets by deeply managing the process of mapping multiple layers was constructed [135]. Inconsistencies between data and function were captured by model testing, including the insufficient total output of ribosomes and RNA polymerases to multiply cell replication, metabolic parameters that were inconsistent with overall growth, and the absence of essential proteins that did not affect cell growth [135]. The discovery of these inconsistent data serves as a new driver to correct model-to-experiment errors, and the development of this model framework is an important step toward whole-cell models. For *S. cerevisiae*, Ye et al. explored the framework of a whole-cell model in which the functions of 1140 essential genes were characterized and associated with five levels of phenotypes, enabling the real-time tracking of the dynamic allocation of intracellular molecules to simulate cell activity [136]. However, due to the simplification of the model framework and the lack of parameters, the model does not extend to all processes in the whole cell.

## 5. Machine Learning in GEMs

Although multiscale GEMs have made important progress in exploring the regulation of metabolic networks, there is still a lack of multilevel resolution schemes to systematically reveal cellular growth and metabolic processes. Machine learning has become an indispensable tool for revealing the regulatory mechanisms of metabolic networks due to its multidimensional data processing capabilities and intelligent analysis strategies [31]. In addition, many machine learning algorithms have been reported to be used in the construction and analysis of multi-scale GEMs. For example, DeepEC uses convolutional neural networks to clarify the enzymatic data in the model [73]; the automatic recommendation tool (ART) and the TeselaGen EVOLVE algorithms explore the effects of transcriptional regulation on target products [74]; machine learning strategies of random forest (RF), elastic network, and neural network improve proteomic utilization of models [75]. Therefore, introducing machine learning into multi-scale GEMs can effectively expand the dimension of the model network and improve the model quality (Figure 3).

### 5.1. Improving the Model Quality

Although various automated tools have been developed to build GEMs, the critical step of filling gaps in models still needs to be managed manually. Recently, the automated metabolic model ensemble-driven elimination of uncertainty with statistical learning (AMMEDEUS) was developed to identify metabolic responses that significantly affect simulation performance [76]. First, growth phenotype data were used to evaluate biomass equations by removing individual genes from the model. K-means was performed to distinguish the ensemble clusters of the model and train the optimized clusters to fill gaps in the simulated growth using a random forest classifier. Finally, the process was performed iteratively until the model could grow under all conditions. A huge dataset of 1000 given models and outstanding computing power for machine learning make AMMEDEUS an important tool to replace the manual management of metabolic networks in models and improve the quality of the automated construction of GEMs [76].

Identifying enzyme committee (EC) numbers is critical for an accurate understanding of enzyme function. However, the reported EC number prediction tools contain computational time and coverage limitations. Therefore, a deep learning-based computational framework was developed for predicting EC numbers from protein sequences (DeepEC) [73]. DeepEC comprises three independent convolutional neural networks and performs three different classification tasks: classifying the input protein sequences as enzyme proteins, modeling in a standard dataset of 387,805 protein sequences, and predicting the third and fourth EC numbers [73]. Therefore, DeepEC can automatically predict EC numbers in a high-throughput computing manner, providing a powerful tool for models to precisely define the relationship between genes and proteins.

### 5.2. Improving the Prediction Accuracy

Improving prediction accuracy is a huge challenge for models, and combining machine learning with GEMs can aid in solving this problem. Zhang et al. developed a method for engineering targets identified by GEMs and machine learning-trained screening of high-throughput biosensors to explore optimal synthetic pathways for tryptophan [137]. First, five target genes were identified in a combined library of 7776 genes by GEM, and the promoters controlling these five genes were mined. Next, multiple training rounds were performed on promoter expression data by combining two machine learning methods, namely, ART and TeselaGen EVOLVE, to explore the optimal gene design scheme. Finally, this integrated strategy increased tryptophan titer and productivity by 74% and 43%, respectively [137]. Moreover, in *E. coli* K-12 MG1665, the GEM framework was restricted by the proteomics and fluxomics of 21 strains. The model significantly improved the prediction accuracy of quantitative proteomic data by integrating a comprehensive machine learning strategy of RF, elastic network, and neural network, and its prediction error was reduced by >40% [74].

For *S. cerevisiae*, Culley et al. proposed a multimodal learning framework based on fluxomics and transcriptomics, which utilizes transcriptome data and GEMs to predict the phenotypic characteristics of cells [75]. In particular, three machine learning techniques, including support vector regression (RF) and artificial neural networks (ANNs), were used to analyze high-dimensional omics data and explore the correlations between features to predict cells phenotype [75]. Three machine learning methods, namely, Bayesian efficient multiple-kernel learning (BEMKL), bagged random forest (BRF), and multimodal artificial neural network (MMANN), were used for data integration and multi-view fusion [75]. Importantly, to address the multidimensionality of the data, (1) sparse group lasso (SGL) was used for the resolution of biological function correlations, (2) nondominated sorting genetic algorithm II (NSGA-II) was used for the optimization of multiple objectives, and (3) iterative random forests (iRF) was used to analyze nonlinear interactions between biometrics [75]. Therefore, the specific GEMs of 1.229 strains of *S. cerevisiae* mutants were constructed using this framework. The introduction of mechanoflux signatures significantly increases the range of discernible mechanobiological insights, providing analytical tools for uncovering unknown interactions between biological domains [75].

John et al. used Bayesian inference and linlog kinetics to develop a scalable metabolic ensemble modeling simulation capable of inferring kinetic parameters of large metabolic models with multiomics-scale datasets. This provides a solution for a complete kinetic description in the kinetic model [76]. In addition, multiple machine learning algorithms were integrated with GEMs for a comprehensive exploration of the effects on cellular metabolism, such as regularized multinomial logistic regression (RMLR) [77], primary elementary modal analysis (PEMA) [78], and genetic algorithms (GA) [98,138].

### 5.3. Exploring Metabolic Networks

GEMs are often used to explore synthetic pathways of target products, and clarifying their metabolic network is conducive to specifying a comprehensive metabolic regulation strategy. The introduction of machine learning is conducive to analyzing datasets from multiple perspectives and comprehensively exploring and analyzing the distribution of metabolic fluxes of cells. A regularization optimization framework combining PCA and a stoichiometric flux analysis method, primary metabolic flux pattern analysis (PMFA), was proposed, identifying the flux patterns that explain most flux variations [139]. In addition, a sparse PMFA was developed to interpret linear combinations of reaction activities in principle components, providing insights into the interactions between reactions [139]. Therefore, PMFA identifies six mitochondrial pathways in response to changes in oxygen availability in a genome-wide metabolic network analysis of *S. cerevisiae* and explains their metabolic regulatory mechanisms [139].

An automated procedure based on a two-stage GA was developed to automatically generate hypotheses to explain negative interactions between genes [140]. The program overlays genetic interactions between 185,000 metabolic gene pairs into GEMs and introduces machine learning to reconcile differences between predicted and observed phenotypes to demonstrate genetic interactions in small-molecule metabolism and establish a GEM performance range [140]. This computational model reveals the relationship between mutational effects and genetic interactions and proposes mechanistic hypotheses critical for systematically optimizing the GEM structure [140]. Dynamic fundamental mode regression discriminant analysis (dynEMR-DA) was proposed, which maps flux data into a space defined by dynEM and fits the NPLS-DA model with a discriminant purpose [79]. The focus of this model is to capture dynamic fundamental patterns with large performance differences driven by the environment [79]. This model simplifies the dynamic model and combines experimental data and fluxomics to identify changes in metabolic pathways driven by the environment, which is beneficial for probing small changes in cellular metabolic networks early in the culturing process [79]. Moreover, MFlux (http://mflux.org) was developed based on three machine learning algorithms, support vector machines (SVM), k-nearest neighbors (k-NN), and decision trees, to predict bacterial central metabolism. It utilizes 10,013 bacterial metabolic data and integrates machine learning with GEMs to explore the complex relationship between influencing factors and metabolic fluxes [80]. MFlux can reasonably predict the central metabolic flux distribution of 30 bacteria through different culture conditions [80].

## 6. Conclusions and Perspectives

After two decades of development, GEMs have become an indispensable tool for systematically exploring cell growth and metabolism. With the development of biochemical research and omics technology, GEMs are not only limited to the exploration of the metabolic network but also extended to the gene level [99], protein level [100], and transcription level [58]. GEMs provide theoretical guidance for the design of high-yielding strains, such as 3-hydroxypropionic acid [141], lactic acid [142], isobutanol [143], and provide new insights for the creation of cell factories. Based on this multiexpansion, GEMs are widely used in industry, agriculture, and medicine [5,144,145]. However, it is still a huge challenge to use models to fully simulate the complex metabolic network and actual growth state in cells.

In the future, whole-cell models will eventually become the goal for building models of different strains. Although multiple whole-cell models have emerged, the development of truly fully functional whole-cell models remains a challenge. First, it is difficult to clarify all cell mechanisms and obtain accurate and massive experimental data because, compared to *M. genitalium*, the cellular processes of most industrial microorganisms are extremely complex, and there are many unknown areas. Unclear data and mechanisms make it difficult to build the model. Second, the perfect framework is a key factor in building a model. The cell morphology and life cycle of different strains are different, and a single model framework is not suitable for other strains. Finally, efficient and novel toolkits and powerful computing power are indispensable conditions for constructing and analyzing whole-cell models. Overall, advances in assays and algorithms will facilitate the whole-cell modeling of multiple strains, advancing microbial biological discovery and the comprehensive design of cell factories.

## Figures and Tables

**Figure 1 biomolecules-12-00721-f001:**
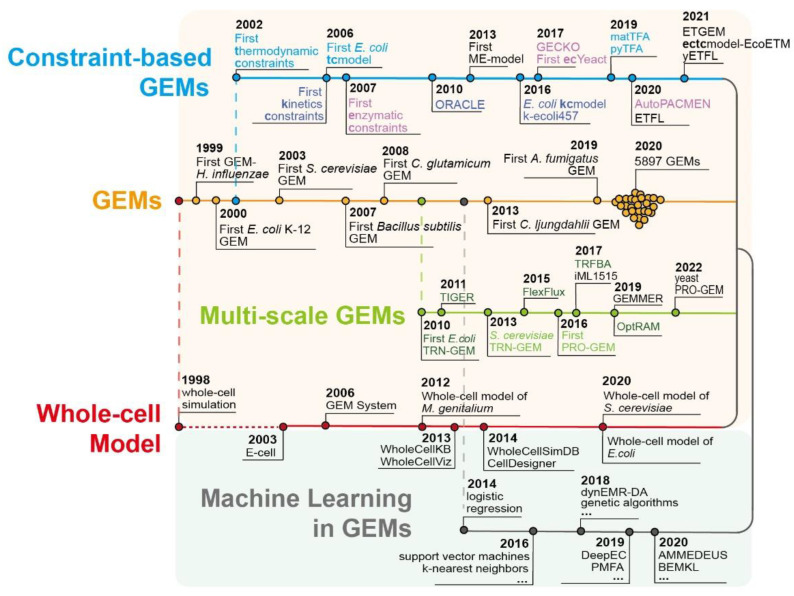
The main development timeline of multiscale GEMs and the application of machine learning.

**Figure 2 biomolecules-12-00721-f002:**
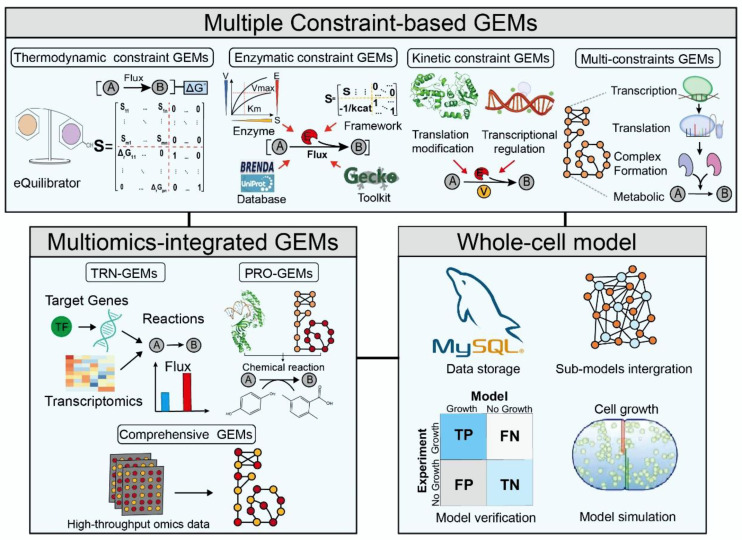
The main classification and construction framework of multiscale GEMs.

**Figure 3 biomolecules-12-00721-f003:**
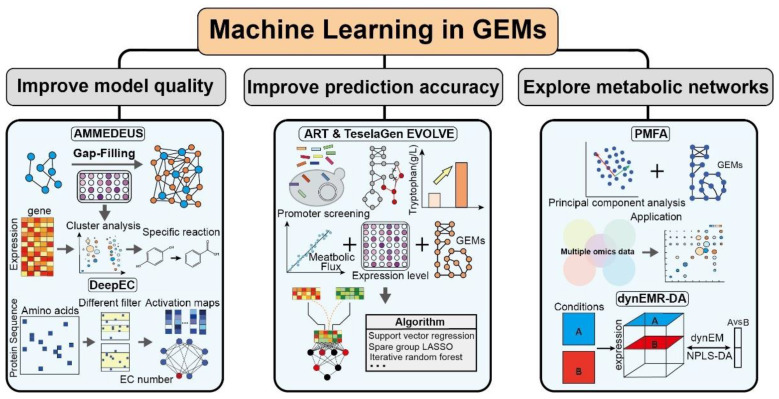
Representative application of machine learning in GEMs.

**Table 1 biomolecules-12-00721-t001:** Algorithms and frameworks for the construction and application of multiscale models.

Model Type	Year	Algorithm/Framework	Language	Task	Reference
Constraint-based models	2007	TMFA	MATLAB	Thermodynamic constraint model	[36]
	2019	MatTFA, pyTFA	MATLAB, Python	Toolkit of build thermodynamic constraint model	[41]
	2007	FBAwMC	MATLAB	Calculation scheme of enzyme concentration	[46]
	2012	MOMENT	MATLAB	Enzymatic constraint model	[47]
	2017	GECKO	MATLAB	Comprehensive framework for enzyme constraint models	[48]
	2006	Structural Kinetic Modeling	MATLAB	Dynamic analysis of metabolic systems	[49]
	2008	MASS framework	MATLAB	Evaluate the dynamic properties of the model and formulate a timescale hierarchy	[50]
	2010	ORACLE	MATLAB	Introducing the state space of the enzyme into the model	[51]
	2008	Ensemble Modelling	MATLAB	Framework for Steady-State kinetics model	[52]
	2016	ABC-GRASP	MATLAB	Framework for modeling uncertain dynamics data	[53]
	2021	ETGEM	Python	Framework of enzyme constraints and thermodynamic constraints	[54]
	2020	Expression and Thermodynamics Flux models	Python	Multi-omics integrated framework	[55]
Multi-scale Integrated models	2011	TIGER	MATLAB	Integrate TRN and GEM platforms	[56]
	2015	FlexFlux	Java	Integrate TRN and GEM platforms	[57]
	2010	Probabilistic Regulation of Metabolism	MATLAB	Toolkit of integrate TRN and GEM	[58]
	2017	TRFBA	MATLAB	Toolkit of integrate TRN and GEM	[59]
	2019	OptRAM	MATLAB	Predict optimal metabolic flux in TRN-integrated GEM	[60]
	2016	GEM-PRO	MATLAB	Integration of protein structure with GEM	[61]
	2019	GEMMER	Python + Java	Database for multiscale modeling	[62]
Whole cell model	2006	GEM System	Java	Toolbox for building metabolic pathways in whole-cell models	[63]
	2021	Pathway Tools	Python + Java	Software for pathway and genetic data	[64]
	2013	WholeCellKB	Python + SQL	Database of whole-cell models	[65]
	2020	CellML	XML	Mathematical models describing cellular physiological systems	[66]
	2003	E-Cell	C++	Multiplatform cell simulation system	[67]
	2014	CellDesigner	SBML	modeling tool for biochemical networks	[68]
	2009	Complex pathway simulator	SBML	Software for biochemical network modeling and simulation	[69]
	2009	Biochemical simulations	Python	Random mixture algorithm	[70]
	2014	WholeCellSimDB	Python + Java	Database of whole-cell model predictions.	[71]
	2013	WholeCellViz	Java + SOL	visualization for whole-cell models	[72]
Machine learning-based models	2019	DeepEC	Python	EC number prediction by deep learning	[73]
	2020	ART, TeselaGen EVOLVE	Python	Multi-level training datasets for accurate prediction	[74]
	2020	BEMKL, bagged random forest, multimodal artificial neural network, sparse group lasso, NSGA-II, iterative random forests	Python	Multiomics and multimodal algorithms to predict phenotypes	[75]
	2020	AMMEDEUS	Python	Tools to identify changes in model structure	[76]
	2014	regularized multinomial logistic regression	MATLAB	Tool for phenotypic inverse prediction of growth conditions	[77]
	2016	primary elementary modal analysis	Python	Identifying metabolic patterns in fluxomics based on metabolic pathways	[78]
	2018	dynEMR-DA	MATLAB	Algorithm for environment-driven dynamic performance discrimination	[79]
	2016	support vector machines, k-nearest neighbors, decision trees	MATLAB	Method for rapid prediction of bacterial heterotrophic fluxomics	[80]

ABC-GRASP: Approximate Bayesian Computation-General Reaction Assembly and Sampling Platform; AMMEDEUS: automated metabolic model ensemble-driven elimination of uncertainty with statistical learning; ART and TeselaGen EVOLVE: Automatic Recommendation Tool and TeselaGen EVOLVE; BEMKL: Bayesian efficient multiple-kernel learning; dynEMR-DA: Dynamic Fundamental Mode Regression Discriminant Analysis; ETGEM: Pyomo-based model framework integrating enzymatic constraints and thermodynamic constraints; FBAwMC: Flux Balance Analysis with Molecular Crowding; GECKO: GEMs with Enzymatic Constraints using Kinetic and Omics data; GEM-PRO: genome-scale model with protein structure; GEMMER: genome-wide tool for multi-scale modeling data extraction and representation; MOMENT: MetabOlic Modeling with ENzyme kineTics; NSGA-II: nondominated sorting genetic algorithm II; OptRAM: optimization of regulatory and metabolic networks; ORACLE: Optimization and Risk Analysis of Complex Living Entities; TMFA: thermodynamically based metabolic flux analysis; TIGER: toolbox for integrating genome-scale metabolism; TRFBA: transcriptional regulation flux balance analysis.

## Data Availability

Not applicable.

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
