# Peer review of "Construction of Multiscale Genome-Scale Metabolic Models: Frameworks and Challenges"

_biomolecules, 2022, doi:10.3390/biom12050721_

Round 1

Reviewer 1 Report

The authors present a review of multi-scale metabolic models for strain design. The topic is timely as models at a single scale are now well-developed and thus integration of multiple systems can be seen as a logical next step. The coverage of topics seems excellent and the organization is logical. I only have a few minor suggestions:

Minor comments

  • The figures in general look quite nice – Figure 2 fonts and the fonts on the right panel of Figure 3 maybe a bit small. The rotation in Figure 2 makes it a bit more difficult to read and a square representation might be preferable
  • Table 1 is massive, and likely still not comprehensive – maybe more appropriate for a supplement, or select representative examples
  • It would seem to me to be better if the machine learning section could better fit under the theme of multi-scale modeling, for example either through choosing inherently multi-scale efforts or applications that enable multi-scale modeling specifically. For example, the PMFA method seems to be a method that exists entirely at the ‘metabolic’ level without any multi-scale aspect. As written, it seems to be a collection of any methods at the intersection of metabolism and machine learning while losing the objective of multi-scale modeling.
  • Although the title and abstract mention biodesign/cell factories, there is little of this content in the manuscript. I would suggest either adding a separate section on multi-scale modeling for metabolic engineering (following this theme), or changing the title/abstract slightly to avoid giving the reader the wrong impression about the content.

Reviewer 2 Report

The manuscript by Bi et al. provides an extensive overview of the state-of-the-art tools and workflows dedicated to the construction and analysis of multi-scale, multi-constrained GEM models taking into account the challenges and perspectives in this research field. This is a review that is definitely useful in the design of cell factories. Overall, it was a manuscript I enjoyed reading. However, despite the overview scale, the manuscript does not contain some essential references considering methodological aspects, tools and some challenges that are not covered in the current version and the inclusion of that with corresponding comments and/or discussion would differ the overview from the majority of similar ones. Thus, there are some issues that need to be addressed before this manuscript can be accepted for publication:

  1. Tools relevant to the overview and not mentioned as well as not discussed in the manuscript:

iMAT- https://doi.org/10.1093/bioinformatics/btq602

MADE - https://doi.org/10.1093/bioinformatics/btq702

INIT - https://dx.doi.org/10.1371%2Fjournal.pcbi.1002518

mCADRE - https://doi.org/10.1186/1752-0509-6-153

ΔFBA - https://doi.org/10.1371/journal.pcbi.1009589

GIM3E - https://doi.org/10.1093/bioinformatics/btt493

multiTFA – https://doi.org/10.1093/bioinformatics/btab151

INTEGRATE - https://doi.org/10.1371/journal.pcbi.1009337

KBASE (https://doi.org/10.1038/nbt.4163), ModelSEED (https://doi.org/10.1093/nar/gkaa746) and CarveMe (https://doi.org/10.1093/nar/gky537) as tools for automated reconstruction of genome-scale metabolic models including gap-filling algorithms.

BioUML - https://doi.org/10.1093/nar/gkac286 (as a platform for the construction and visualization of whole-cell models)

  1. I would also highly recommend to add the description of methods and tools for the dynamic flux balance analysis.
  2. MEMOTE (https://doi.org/10.1038/s41587-020-0446-y) and FROG (https://www.ebi.ac.uk/biomodels/curation/fbc) as a community attempts to standardize and improve the GEM models quality.
  3. Overviews on the topic: Zielinski et al., 2020 (https://doi.org/10.3390/microorganisms8122050) , Panikov, 2021(https://doi.org/10.3390/microorganisms9112352), Kim et al., 2021 (https://doi.org/10.1016/j.coisb.2021.03.001) , Lu et al., 2021 (https://doi.org/10.1016/j.tibtech.2021.06.010), Chen&Nielsen, 2021 (https://doi.org/10.1016/j.coisb.2021.03.003)

Minor comments:

  1. Table 1: The order of presented methods is unclear: chronology of the origination (however, it is not according to the column Year), from simple to complex or what?
  2. “Multi-constrained model” is a mostly used term.
  3. Line 580, “Firstly,…”
  4. Line 621, “…a sparse PMFA…”
  5. Line 651, “…also extended…”

Round 2

Reviewer 2 Report

The authors satisfactorily addressed my comments.